# What factors are important to whom in what context, when adults are prescribed hearing aids for hearing loss? A realist review protocol

Emma Broome [1,2] Carly Meyer,[3] Paige Church,[1,4] Helen Henshaw [1,2]

¹National Institute for Health and Care Research (NIHR), Nottingham Biomedical Research Centre, Nottingham, UK
²Hearing Sciences, Mental Health and Clinical Neurosciences, School of Medicine, University of Nottingham, Nottingham, UK
³Department of Clinical, Educational and Health Psychology, Centre for Behaviour Change, University College London, London, UK
⁴NIHR Clinical Research Network (CRN) East Midlands, Nottingham Health Science Partners, Nottingham, UK

**Correspondence to**
Emma Broome;
emma.broome1@nottingham.ac.uk

## ABSTRACT

**Introduction** Hearing aids are the gold standard treatment to help manage hearing loss. However, not everyone who needs them has them, and of those who do, a significant proportion of people do not use them at all, or use them infrequently. Despite literature reviews listing key barriers and enablers to the uptake and use of hearing aids, there is little evidence to describe how this varies by population and context. This review will describe *what factors are important to whom in what context* when considering the provision of hearing aids for hearing loss in adults.

**Methods and analysis** The aims of this review are as follows: (1) To iteratively review and synthesise evidence surrounding the provision of hearing aids for hearing loss in adults. (2) To generate a theory-driven understanding of factors that are important, for whom, and in what context. (3) To develop a programme theory describing contexts that can support the provision of hearing aids to result in improved outcomes for adults with hearing loss. A scoping literature search will aid the development of programme theories, to explain how the intervention is expect to work, for whom, in what circumstances and in which contexts. We will locate evidence in the following databases: CINAHL, Cochrane Library, EMBASE, MEDLINE, PsycINFO, PubMED, Web of Science with no date restrictions. A realist analytic approach will be used to refute and refine these initial programme theories. Throughout the review, relevant key stakeholders (eg, patients and clinicians) will be consulted to test and refine the programme theories.

**Ethics and dissemination** This study was approved by the University of Nottingham Faculty of Medicine and Health Sciences Research Ethics Committee: (FMHS 95-0820) and the London Brent NHS Research Ethics Committee (Ref: 21/PR/0259). The review will be reported according to the RAMESES guidelines and published in a peer-reviewed journal.

**PROSPERO registration number** CRD42021282049.

## INTRODUCTION

Hearing loss represents a major public health concern and a growing disease burden in an ageing global population, with the vast majority of losses attributed to age-related sensorineural degeneration.[1] Globally, over 1.5 billion people live with hearing loss, and

## STRENGTHS AND LIMITATIONS OF THIS STUDY

⇒ The review team is interdisciplinary including expertise in adult aural rehabilitation, clinical audiology service provision and health behaviour change, which will support the development of robust programme theories across different disciplines.
⇒ Using a realist approach to identify context, mechanism and outcome configurations will further our understanding of why the provision of hearing aids results in different outcomes across adults with hearing loss.
⇒ The review may be limited by the richness and relevance of evidence relating to mechanisms of change and contextual elements available in the literature.

this is expected to rise to 2.5 billion by 2050. In the UK, hearing loss affects 1 in 6 of the population, and it is estimated that by 2035, more than 15 million people, or 1 in 5 of the UK population, will be affected.[2 3]

More than 40% of people aged over 50 years old live with hearing loss, increasing to more than 70% of those aged 70 years or older.[2 3] Hearing is the sense most relied on to communicate and engage with others.[1] In 2019, hearing loss was the third largest source of global years lived with disability (YLD), and the leading source of YLD for adults older than 70 years of age.[4] Hearing loss is something that almost every one of us will eventually experience.[5] However, hearing loss is not a benign consequence of ageing.[6] It is a multifactorial condition, influenced by; genetic factors that determine the rate of degeneration, pre-existing ear conditions, chronic illnesses, noise exposure, the use of ototoxic medications, lifestyle factors,[1] and socioeconomic inequalities.[7]

In the absence of medical or surgical treatments for sensorineural hearing loss,[8] the current gold-standard management option is the provision of hearing aids to amplify sounds.[9] An estimated 6.7 million people in the UK alone

could benefit from hearing aids, but only around 2 million people have them.[3 10] Of those who do have hearing aids, recent data show that 20% do not use them at all, and 30% use them infrequently.[11] Taken together, this represents a huge unmet hearing healthcare need.

The impacts of untreated hearing loss in adults are far-reaching and include problems with listening and communication,[9] reduced satisfaction with relationships,[12] increased loneliness and social isolation,[13 14] reduced employment and economic productivity,[15] poorer mental health[16] and poorer quality of life.[17] Known associated hearing loss comorbidities include; cardiovascular disease,[18] stroke,[19] diabetes[20] and cognitive impairment and dementia.[21] Estimated risk of dementia for those with untreated hearing loss is twice as likely for those with mild hearing loss compared with those without hearing loss, three times greater for those with moderate hearing loss and five times greater for those with severe hearing loss.[21] In both the initial (2017) and updated (2020) Lancet Commission Reports on dementia prevention,[22 23] hearing loss was identified as the largest modifiable mid-life dementia risk factor, and that intervention for hearing loss could prevent or delay up to 8% of dementia cases. Finally, evidence suggests a twofold to threefold increase in the number of falls experienced by adults with hearing loss,[24–26] but not for those who used hearing aids.[26] Hearing aids are an effective intervention for the ongoing management of mild–moderate hearing loss in adults and its' associated difficulties, with benefits including improved communication, better hearing-specific and health-related quality of life, increased economic prospects, reduced loneliness and better mental health.[8 9 27] The earlier hearing aids are provided and used, the better the outcomes for the individual.[28] National Institute for Health and Care Excellence (NICE) guidance states that hearing aids should be offered to adults whose hearing loss affects their ability to communicate and hear, and that two hearing aids should be offered to those with aidable hearing loss in both ears.[29] However, to what degree an individual with hearing loss seeks, is eligible for, adapts to, uses and benefits from hearing aid(s) is influenced by a myriad of factors spanning personal, interpersonal, environmental, psychosocial, societal, cultural and policy origins.[29]

Due to its typically gradual onset, recognition of acquired hearing loss can be poor among those affected, which can result in a delay in help seeking of up to 10 years.[27] Data from a longitudinal cohort study in the USA show the average time from hearing aid candidacy to hearing aid fitting is 8.9 years.[30] In the UK, adults access National Health Service (NHS) hearing services via general practice.[31] Yet, it was reported in 2011 that 45% of those presenting to their general practitioner with hearing difficulties had failed to get an onward referral for audiological assessment.[32]

Once referred for hearing assessment, a clinical decision will be made regarding eligibility for hearing aid provision. For those who are eligible, a diverse range of barriers to hearing aid use, benefit and satisfaction have been reported in the literature, including discomfort, sound quality, perceived benefit, background noise, sound of own voice, ear infections, wax, manual dexterity problems, perceptions of the clinician or service, patient–clinician rapport, peer support, perceived stigma and self-stigma.[33–37] However, it is vital to acknowledge that different barriers will apply to different individuals at different times, and this will vary depending on the specific context.

In 2015, the UK Action Plan on Hearing Loss set out five key objectives to tackle the rising prevalence and personal, social and economic costs of uncorrected hearing loss. It provides a national plan to spread good practice and address variation in access and quality of services experienced by people with hearing loss. The key objectives are as follows: early diagnosis, integrated patient-centred management, ensuring those diagnosed do not need unscheduled care or become isolated and ability to partake in everyday activities including work.[38] To help achieve these objectives, a fundamental building block is to understand the complexities of routine hearing aid provision for adults with hearing loss, which accounts for the vast majority of activities undertaken within NHS adult hearing services.

Realist review is a theory-driven approach to reviewing and synthesising evidence to understand complex interventions.[39] Realist review is philosophically based in realism, incorporating both positivist and social constructivist approaches, recognising that social systems influence objective knowledge.[40] Realists posit that the world is stratified, and that complex mechanisms, residing in the world of the real, are hidden.[41] At present, there is little or no understanding of how and why hearing aid provision results in different outcomes for patients. Existing reviews have focused on lists of barriers to hearing aid use and have not tended to address the underlying mechanisms by which context influences outcomes of hearing aid provision.[37] The purpose of this realist review is to examine what factors are important to whom in what context regarding the provision of hearing aids for hearing loss in adults. A realist approach accounts for the inherent complexity in interventions such as hearing aid provision, moreover it recognises that context influences human behaviour. A realist explanatory approach addresses a key gap in the literature by exploring how and why there are interactions between context and outcomes for patients who are provided with a hearing aid. Therefore, furthering our understanding about what currently works well, and where improvements could be made.

## METHODS AND ANALYSIS
### Review question/aim
What factors are important to whom in what context, when adults are prescribed hearing aids for hearing loss?

### Objectives
1. To iteratively review and synthesise evidence surrounding the provision of hearing aids for hearing loss in adults.

2. To generate a theory-driven understanding of factors that are important, for whom, and in what context.
3. To develop programme theories describing contexts that can support the provision of hearing aids to result in improved outcomes for adults with hearing loss.

## Patient and public involvement

People with hearing loss, their family and clinicians helped define the topic of this review via a James Lind Alliance priority setting partnership,[42] priority question #4: *What are the reasons for low hearing aid uptake, use and adherence?*

This protocol has been reviewed by a Patient Research Partner who has lived experience of managing hearing loss with hearing aids, for content, clarity and relevance.

## Study design

This realist review will formulate, test and refine programme theories to generate new insights into factors important for the provision of hearing aids for hearing loss in adults.

A programme theory is an overarching theory which explains *how* an intervention is expected to work,[43] for *whom*, in what *circumstances* and in which *contexts*.[44] Initial programme theories describe causal relationships with the purpose of the realist review to look for recurring patterns to explain what it is about an intervention in a given context that makes it work (or not). This review will use realist explanatory theory to explore how the provision of hearing aids results in different outcomes for different people. Realist programme theories can be expressed as statements comprised of contexts, mechanisms and outcomes. Contexts (C) can be defined as social and environmental factors in the backdrop of the intervention.[45] Mechanisms (M) as causal factors which bring about changes through the implementation of an intervention.[46] Finally, intervention outcomes (O), which can either be intended or unintended.[47] Context–mechanism–outcome configurations can be used to produce causative explanations for how the intervention works.[47] An example programme theory is outlined below:

*If an audiologist discusses, describes and explains the extent and implications of a patients hearing loss during the hearing assessment appointment and offers the patient an opportunity to ask questions, then the patient has an increased understanding of their hearing loss and how it affects them personally which results in greater acceptance of their hearing loss and an increased readiness to address their hearing loss through the use of hearing aids.*

The review started in September 2021 and the anticipated end-date is no later than end May 2023. It will follow Pawson's five stages in conducting realist reviews[39] and is summarised in table 1.

## Step 1: Clarify the scope

A scoping search will be conducted to locate and generate initial programme theories which relate to the provision of hearing aids for adults with hearing loss. This process will be two-fold:

1. An exploratory informal literature search of explanatory theories which may be relevant to explaining how the intervention works.

**Table 1** Adapted from Pawson, Greenhalgh[39]

| Steps in conducting a realist review | |
|---|---|
| 1. Clarify the scope | ▶ Locate existing theories through informal searching and input from stakeholders<br>▶ Generate and articulate the key programme theories to be explored in the review |
| 2. Search for evidence | ▶ Pilot and refine search strategy with input from an information specialist<br>▶ Search electronic databases, interrogate reference lists and search grey literature<br>▶ Screen and select documents based on extent to which they test or develop the programme theories from step 1 |
| 3. Data extraction and organisation | ▶ Extract relevant data from sources and organise in bespoke data extraction form |
| 4. Synthesise evidence and draw conclusions | ▶ Search for patterns in the data<br>▶ Use realist analysis to develop Context–Mechanism–Outcome configurations<br>▶ Refine initial programme theories |
| 5. Dissemination | ▶ Findings presented in narrative form in line with the RAMESES publication standards[55] |

2. Reviewing primary data exploring views and experiences of barriers and facilitators to using hearing aids from focus groups and semistructured interviews with relevant stakeholders (eg, adults with hearing loss and audiologists) previously conducted by the research team.

The scope will include the provision of hearing aids in adult auditory rehabilitation and will include wider relevant literature on personal factors (eg, motivation) and social factors (eg, peer support) which may influence which elements works well within hearing aid provision.

Documents sourced within the scoping search will be examined for evidence which will be used to develop initial programme theories of the provision of hearing aids to adults with hearing loss and thus explain how hearing aid provision is supposed to result in its intended outcome (eg, hearing aid use). The initial programme theories developed from the scoping search will be revised by members of the research team who have knowledge of the field of adult auditory rehabilitation including academics, audiologists and public involvement representatives. The initial programme theories will be then tested, developed and refined against data from documents included in the review.

## Step 2: Search for evidence

To identify suitable evidence to test and refine the initial programme theories developed in step 1.

### Search strategy

Following the convention of a realist review, the search strategy will be purposive and will include the following elements:

1. Search terms will be entered into the following electronic databases: CINAHL, Cochrane Library,

**Box 1    Example search terms**

Search terms in Medline (OVID)
1. exp Hearing Loss/
2. exp Deafness/
3. exp Persons With Hearing Impairments/
4. exp Presbycusis/
5. (hearing loss* or deaf* or hearing impair* or hearing disabilit* or hearing disorder* or hearing handicap* or hearing problem* or presbycus* or presbyacus* or auditory rehabilit*).af.
6. exp HearingAids/
7. (hearing aid* or listening device* or sound amplif* or acoustic amplif* or hearing device*).af.
8. (treatment adherence and compliance).mp. [mp=title, abstract, original title, name of substance word, subject heading word, floating sub-heading word, keyword heading word, organism supplementary concept) word, protocol supplementary concept) word, rare disease supplementary concept) word, unique identifier, synonyms]
9. exp Patient Compliance/
10. (prescri* or provi* or complian* or cooperat* or co operat* or non complian* or noncomplian* or non adheren* or nonadheren* or accept* or nonaccept* or satisfaction or benefit* or adapt* or perception* or use* or usage or adopt* or uptake* or reject* or return* or success* or orientat* or take-up or utilis* or non-use).af.
11. 1 or 2 or 3 or 4 or 5
12. 6 or 7
13. 8 or 9 or 10
14. 11 and 12
15. 13 and 14
16. 15 and 'Adult'.sa_suba.

EMBASE, MEDLINE, PsycINFO, PubMed and Web of Science, Google Scholar and NICE Evidence search or equivalent. The list of search terms will be piloted and modified with an information specialist. An example of the search terms for PubMed is given in box 1. The full search strategy for all databases, including any filters and limits used is outlined in online supplemental material S1.

2. Interrogation of reference lists of relevant reviews and primary studies, with forward and backward citation tracking.
3. Searching grey literature including policy documents, charities, user groups and patient associations.

There will be no date restrictions for the formal search[48] or geographical restrictions.

The search terms were developed in collaboration with information specialists at the University of Nottingham Libraries.

Initial searches were conducted March 2022. As the review progresses additional searches will be undertaken, as required, throughout the synthesis. Additional data will be used to test the programme theories until theoretical saturation has been achieved.[39]

## Screening and selection of studies

Document selection will be based on the extent to which identified evidence can contribute towards testing and development of the initial programme theories developed in step 1.

Inclusion criteria:
► Adults provided with hearing aids for the primary complain of hearing loss.
► Any healthcare practitioners, for example audiologists, supporting adults provided with hearing aids for hearing loss.
► Communication partners, for example family members, supporting adults provided with hearing aids for hearing loss.
► Adult audiology patient pathway including assessment and auditory rehabilitation.
► Any study design. Documents such as editorials, opinion pieces, commentaries, process evaluations, qualitative research, programme manuals and systematic reviews may be included, if holding information relevant to developing a programme theory.
► Any outcomes related to the use of hearing aids by adults with hearing loss.

Exclusion criteria:
► Participants <18 years old.
► Documents which do not summarise an empirical study.
► Non-English papers.

Documents identified during the search strategy will be downloaded into Endnote reference management software where duplicate returns will be removed, then returns will be imported into Covidence review management software for document screening, and data extraction.

Selection of relevant evidence will be systematic and will follow a two-step procedure. The titles and abstracts will first be independently screened by two reviewers to assess eligibility for inclusion. Subsequently, the full text of potentially relevant documents will also be independently screened by two reviewers to assess against eligibility criteria for inclusion in the review. The full text will be obtained for documents where the title and/or abstract is too vague to clarify whether they meet the inclusion criteria. The following identification tool, comprised of three questions, will be used to assist with the screening of evidence:[49]

1. Does the document summarise an empirical study of hearing aids for adults with hearing loss?
2. Does the document indicate the specific context of adult auditory rehabilitation?
3. Does the document provide data on patient outcomes?

A list of reasons for exclusions will be recorded by the reviewers. Disagreements will be discussed by both reviewers until a consensus is reached. The wider research team will adjudicate any contested evidence if required.

The selection criteria listed below will be applied by the reviewers to the full text of all potentially relevant documents. There will be no restrictions to types of study design eligible for inclusion in keeping with realist review guidelines.[44] As the screening and selection progresses, the inclusion and exclusion criteria may be iterated.

Selection tool adapted from the following:[49]

1. Does the full-text document still indicate adult audiology patient pathway?
2. Does the full-text document describe the setting (eg, primary care practice, hospitals, audiology services, outpatient care, social care community)?
3. Does the full-text document indicate empirical research (eg, includes a description of methods, data collection and analysis)?
4. Does the full-text document describe a population who are provided hearing aids for hearing loss as the primary complaint (not secondary to provision, eg, tinnitus)?
5. Does the full-text document describe any outcomes related to the use of hearing aids by adults with hearing loss (eg, uptake, use, adherence, acclimatisation, acceptance, benefit, satisfaction and maintenance)?
6. Does the full-text document describe the processes or context related to the provision of hearing aids for adults with hearing loss (or is there a reference to the process/context in a companion document)?

Document screening and selection will be recorded using an adapted Preferred Reporting Items for Systematic Reviews and Meta-Analyses flow diagram.[50]

### Appraisal of included articles

During the data extraction phase, the full text of the documents will be screened for their relevance to the initial programme theories, and their rigour, using the definitions below:[51]

► Relevance: whether the study contributes to the development of the programme theories under examination.
► Rigour: whether the methods used to draw inferences are credible and trustworthy to test a particular theory in terms of sample size, data collection and analysis and inferences drawn by the authors.[51]

The realist approach does not adhere to the hierarchy of study designs used in other types of review for example, systematic reviews. In realist review, diverse data sources will be used to further our understanding of how and why programmes function.[52] Data from the included documents will be assessed by their own merit, rather than considering the document content as a whole, for example, there may be causal insights which could contribute towards the development of the programme theories in seemingly methodologically weak studies.[53] Similarly, a document which meets the selection criteria may not contain any relevant data which aids the refinement of the programme theories.

The quality appraisal process involves assessing each document on a case-by-case basis for the concepts of *relevance* and *rigour*. The following process will be employed:
1. Reading the full-text document in its entirety.
2. Documenting what contribution the document makes to the synthesis (relevance) for example, does the document contain content which can be examined against the initial programme theories? Does the document contain information on how the programme works?

3. Documenting any methodological or conceptual difficulties with the document (rigour) for example, appraising the quantitative or qualitative methods used.

Relevance and rigour characteristics of included studies will be recorded in a bespoke data extraction form.

A 10% random sample of included study documents will be selected, assessed and discussed regarding quality appraisal, with a second reviewer for consistency. Any disagreements or ambiguity in relevance or rigour will be resolved by discussion with the wider review team to enhance validity and consistency.

### Step 3: Data extraction and organisation

Data extraction for selected documents will follow a hybrid approach and will be undertaken by the lead reviewer, and a second reviewer. This approach to data extraction includes manually theory-tracking by highlighting, annotating and note-taking as well as the use of a bespoke data extraction form to provide a descriptive overview of the evidence.[39 54]

The bespoke data extraction form will be developed, piloted with a small sample of included documents (eg, three, representing peer-reviewed and grey literature), and used to record pertinent characteristics of each included document. As the review progresses and the initial programme theories are iteratively refined, the data extraction form will be modified to ensure all new insights which are relevant to the programme theories are captured. As the data extraction form is modified, included documents will be revisited to ensure that all appropriate and relevant data is extracted. The data extraction form will record the following information:
► Document information: authors, publication date, document type, where and how the document was sourced.
► Context: geographical location, healthcare system context.
► Design: research aim, design, methods, setting, intervention details (type of intervention or programme, participant details, who was involved in the intervention), method of assessment, and summary of reported outcomes.
► Relevance to the review question.

It is anticipated that not all of these aspects will be available for each document.

The theory-tracking phase of data extraction will be guided by the initial programme theories developed in step 1. Selected texts will be imported into a data management software and relevant sections of the document text will be reviewed, highlighted and annotated in relation to contexts, mechanisms and outcomes and/or their relationships. In this phase, any information which relates to theories on how or why the intervention works or does not work will be examined and recorded.

Codes will cover concepts which are considered important and relevant to the initial programme theories. Coding of the data will be both inductive (ie, created during analyses of the data) and deductive (ie, created

from the initial programme theories, identifying data that either confirms or refutes the theory under examination); both approaches will be used to identify contexts, mechanisms and outcomes and the associations between them. The data will be used to refine the initial programme theories, into configurations of contexts and main mechanisms which influence specific outcomes for different populations. As the theories are refined, the included studies will be revisited to search for data relevant to the revised programme theories.

### Step 4: Synthesise evidence and draw conclusions
Step 4 will use a realist analytic approach to build on the evidence, refute or provide alternative explanations for key factors associated with hearing aid provision for adults with hearing loss. The analysis will focus on refining the programme theories to explore what works well in hearing aid provision, how different elements are thought to have made this happen and what needs to be in place for this to occur. Again, this process will be iterative, interrogating and refining the initial programme theories[39] using relevant empirical findings in included documents. Relevant data from each document will be systematically considered to test and refine each programme theory. The extracted codes will be synthesised, moving within and across each document looking for patterns and associated mechanisms and contexts; putting together revised Context–Mechanism–Outcome configurations.

The following realist synthesis strategies will be employed:
► Organising extracted data from the included documents.
► Identifying similarities and juxtapositions across each data source aligned to the realist concepts of context, mechanism, and outcome.
► Examining and exploring disparities between the data.
► Moving within and across each data set, linking demiregularities (patterns) in the data to test and refine each programme theory.
► As the synthesis progresses, it may be necessary to conduct additional iterative literature searches, in order to examine or explore particular aspects of the refined programme theory. Any additional documents identified will provide evidence to support, refute or refine the programme theory.[55]

The data synthesis process will involve reflection and discussion with the research team. Stakeholders, identified through professional contacts of the research team, including healthcare professionals, people with hearing loss and those who they communicate regularly with will be consulted and will assist in the refinement of the final programme theory through workshop discussions.[56] The review team will examine the results of the analysis and assist with interpretation of the findings and refine the analyses in order to progress the final programme theories.

### Step 5: Ethics and dissemination
Ethical approval for the primary data component of our review was granted by both the University of Nottingham Faculty of Medicine (Ref: FMHS 95-0820) and the London Brent NHS Research Ethics Committee (Ref: 21/PR/0259).

Findings will be presented in narrative form and will include tables and figures, with a diagram used to present the final programme theories. The review will be reported in line with the RAMESES standards.[55] We will disseminate through conference, presentations and publications in peer-reviewed journals and in social media publications and blogs that are accessible to patients and the public.

## DISCUSSION
This realist review will provide a theory-driven understanding of what works in terms of the provision of hearing aids for hearing loss in adults. Data generated in this review will provide new insights into causal mechanisms which influence hearing aid provision and outcomes for adults with hearing loss, to further our understanding about how adult hearing healthcare and hearing services may be improved in future.

**Acknowledgements** We thank Research Librarian, Alison Ashmore, from the University of Nottingham Libraries whose constructive comments/suggestions helped to improve the search strategy.

**Contributors** EB and HH conceptualised the study. EB and HH wrote the first draft of the manuscript. Critical review and refinement of the manuscript were provided by CM and PC. All authors have read and approved the final version.

**Funding** This study is funded by the National Institute for Health Research (NIHR) (CDF-2018-11-ST2-016). The views expressed are those of the author(s) and not necessarily those of the NIHR or the Department of Health and Social Care.

**Competing interests** None declared.

**Patient and public involvement** Patients and/or the public were involved in the design, or conduct, or reporting or dissemination plans of this research. Refer to the Methods section for further details.

**Patient consent for publication** Not required.

**Provenance and peer review** Not commissioned; externally peer reviewed.

**ORCID iDs**
Emma Broome http://orcid.org/0000-0002-1607-0594
Helen Henshaw http://orcid.org/0000-0002-0547-4403

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
