## [Reviewer comments · BMJ Open]

ARTICLE DETAILS

TITLE (PROVISIONAL)	What factors are important to whom in what context, when adults are prescribed hearing aids for hearing loss? A realist review protocol.
AUTHORS	BROOME, EMMA; Meyer, Carly; Church, Paige; Henshaw, Helen

VERSION 1 – REVIEW

REVIEWER	Humes, Larry Indiana University Bloomington, Speech & Hearing Sci
REVIEW RETURNED	06-Jan-2022

GENERAL COMMENTS	I found this research proposal to be clearly written and the proposed research well justified. The research plans were novel and appropriate for the problem being addressed. When completed, the study should offer new insights into the factors impeding hearing-aid uptake and use. There were just two issues, both minor. I was confused on Page 10 of the document, Line 6, where it was noted that the review of the prior reviews would be conducted by "a second reviewer" for a 10% random selection of studies. Prior to this, in several locations, two reviewers were already involved. I BELIEVE, then, that it is a 3rd reviewer who will review a 10% sampling of the results from those other two primary reviewers. Also, I could not find a timeline, Gantt chart, etc., that indicated the dates anticipated for each step in the design to be initiated and completed. It is my understanding that such information is typically required for study protocols submitted for review.
--

REVIEWER	Granberg, Sarah Univ Orebro, School of health sciences
REVIEW RETURNED	27-Jan-2022

GENERAL COMMENTS	Thank you very much for a nice manuscript. The authors address an extremely important research question and have chosen a very relevant methodology to investigate the topic. I have not previously located any realist reviews in this area which makes both the topic and the methodology very interesting and original. However, the protocol needs to be clarified in certain aspects. I have used the review function in Word to facilitated the understanding of my comments, see attachment – contact the publisher to view.
--

REVIEWER	Dillier, Norbert University Hospital Zurich, ENT Department, Experimental Audiology
REVIEW RETURNED	01-Feb-2022

GENERAL COMMENTS	The study protocol is carefully prepared and follows rational and structured guidelines. The topic of the proposed study is relevant and the methods seem adequate. My only remaining question relates to the planned schedule of this process. There are a number of iterations described in the protocol which apparently render accurate time estimates difficult. Nevertheless, it would be interesting for the readers to know approximately which timeline and overall effort would be required. When can results of the study be expected?
---

VERSION 1 – AUTHOR RESPONSE

Review 1 comments:

1. There were just two issues, both minor. I was confused on Page 10 of the document, Line 6, where it was noted that the review of the prior reviews would be conducted by "a second reviewer" for a 10% random selection of studies. Prior to this, in several locations, two reviewers were already involved. I BELIEVE, then, that it is a 3rd reviewer who will review a 10% sampling of the results from those other two primary reviewers.

Author response: Thank you for your comments. On Page 10, text has been added to clarify that a second reviewer will assess 10% of included study documents with regard to the quality appraisal of included documents (assessing quality in terms of relevance and rigour). Two reviewers will conduct all other screening activity (titles, abstracts, full text). Please see text added below:

A 10% random sample of included study documents will be selected, assessed, and discussed regarding quality appraisal, with a second reviewer for consistency.

2. Also, I could not find a timeline, Gantt chart, etc., that indicated the dates anticipated for each step in the design to be initiated and completed. It is my understanding that such information is typically required for study protocols submitted for review.

Author response: As realist reviews are an iterative process, we cannot state which stage of the review will happen when, as this type of review takes an iterative approach, with each step overlapping as theories are refined and/or new evidence emerges. However, in light of your comment, we have added additional text to clarify when the review started and the anticipated end date. Please see text below:

The review started in September 2021 and the anticipated end-date is no later than end May 2023.

Reviewer 2 comments:

1. Reference to realist review theory and methodology is missing here.

Author response: We have added the appropriate reference into the text. See below:

Pawson, R., et al., Realist review--a new method of systematic review designed for complex policy interventions. J Health Serv Res Policy, 2005. 10 Suppl 1: p. 21-34.

2. Perhaps add a sentence about the theoretical foundation for this review methodology? Where is it's philosophical roots and what assumptions is it based on?

Author response: We have added the following text to the paragraph which introduces the theoretical foundation for realist review. Please see text added below:

Realist review is philosophically based in realism, incorporating both positivist and social constructivist approaches, recognising that social systems influence objective knowledge [41]. Realists posit that the world is stratified, and that complex mechanisms, residing in the world of the real, are hidden [42].

3. What patient? Characteristics? And does one view from one patient play a significant role?

Author response: Thank-you for this comment. As per the journal guidelines we have included a statement on Patient and Public involvement. We have added text to clarify that the protocol has been reviewed by a Patient Research Partner who has lived experience of managing hearing loss with hearing aids. We acknowledge that one patient does not speak for all, but the review question is aligned to priority question 4 of a James Lind Alliance priority setting partnership, which incorporated the views of patients, clinicians, and other key stakeholders. Please see amended text below:

This protocol has been reviewed by a Patient Research Partner who has lived experience of managing hearing loss with hearing aids, patient for content, clarity and relevance.

4. As stated in the journal guidelines, the dates of the study should be included in the manuscript. Perhaps in this section?

Author response: As realist reviews are an iterative process, we cannot state which stage of the review will happen when, as this type of review takes an iterative approach, with each step overlapping as theories are refined and/or new evidence emerges. However, in light of your comment, we have added additional text to clarify when the review started and the anticipated end date. Please see text below:

The review protocol is registered with PROSPERO (282049) it started in September 2021 and will end by May 2023.

5. I have a comment about “program theory” below. You really need to elaborate a bit about this term. Perhaps here, perhaps below, your decision. From my point of view, this might be tricky to investigate given the methodology. Perhaps use the term “interfered” och “influenced” (Greenhalgh et al., 2011) instead of causal factors? It seems to be different “schools” here, but when I use the term causality, I feel that I really must be sure that $C+M=O$. Given that you do not make any quality appraisal of the included papers, the causal relationship might be weak in some of the included papers.

Author response: Generative causation in realist review refers to underpinning generative forces that cause things to happen in a certain context. These underpinning mechanisms are not easily observable or measurable. To give an example, a seed in soil needs to have the right conditions e.g. water level, pH balance to be able to grow. A realist review seeks to unearth causal mechanisms and further knowledge by positing how and why they are activated in certain circumstances. Although we take on board your comments about causality, we feel that to adhere to our theoretical standpoint and realist review methodology we must keep the wording as it stands.

Considering your comment, we have added additional text to elaborate the term programme theory. Please see additional text below:

Initial programme theories describe causal relationships with the purpose of the realist review to look for recurring patterns to explain what it is about an intervention in a given context that makes it work (or not). This review will use realist explanatory theory to explore how the provision of hearing aids results in different outcomes for different people.

We have also added in an example programme theory. See additional text below:

An example programme theory is outlined below:

If an audiologist discusses, describes and explains the extent and implications of a patient's hearing loss during the hearing assessment appointment and offers the patient an opportunity to ask questions, then the patient has an increased understanding of their hearing loss and how it affects them personally which results in greater acceptance of their hearing loss and an increased readiness to address their hearing loss through the use of hearing aids.

6. In line with the previous comment, "associations" instead?

Author response: Please see comment directly above regarding generative causation in realist review methodology.

7. The term "program theories" needs to be explained further to facilitate the understanding of this protocol. I suggest e.g., to use the definition by your ref no. 51 (Rycraft- Malone) p.3. and to elaborate this definition in relation to hearing aid fitting. In its current form, it does not really make any sense. Being an audiologist myself, I cannot understand what you are trying to identify. You can, of course, use this term but you need to clarify to the reader what you expect to find. If I would have conducted this review, I would have stated three research questions (also in relation to your title) to clarify the scope (also to enhance the structure of this review). 1) For whom are HA-interventions effective? 2) Under what conditions are HA-interventions successful? 3) In which context are these interventions successful? These research questions also align with the conducted searches that you present below. This is, of course, only suggestions to improve the protocol. In its current form, Step 1 is difficult to grasp.

Author response: We have addressed the comment regarding the explanation of programme theories by adding the following text:

Initial programme theories describe causal relationships with the purpose of the realist review to look for recurring patterns to explain what it is about an intervention in a given context that makes it work (or not). This review will use realist explanatory theory to explore how the provision of hearing aids results in different outcomes for different people.

We have also added an example programme theory. See text added below:

An example programme theory is outlined below:

If an audiologist discusses, describes and explains the extent and implications of a patient's hearing loss during the hearing assessment appointment and offers the patient an opportunity to ask questions, then the patient has an increased understanding of their hearing loss and how it affects them personally which results in greater acceptance of their hearing loss and an increased readiness to address their hearing loss through the use of hearing aids.

8. Do you have access to this kind of data, or will you conduct focus groups and interviews? I suggest that you expand the information a bit about ii).

Author response: A part of our larger body of work looking at barriers and facilitators of hearing aid use, we have conducted focus groups with i) people with hearing loss and their communication partners and ii) audiologists working within the National Health Service (UK). We have also conducted in depth qualitative semi-structured interviews with people with hearing loss. These qualitative methods explored the views and experiences of barriers and facilitators to the use of hearing aids. We will use this primary data to generate the initial programme theories to be used in this review. Please see text added below to clarify:

ii) reviewing primary data exploring views and experiences of barriers and facilitators to using hearing aids from focus groups and semi-structured interviews with relevant stakeholders (e.g. adults with hearing loss and audiologists) previously conducted by the research team

9. Inclusion/exclusion criteria is missing in step 2

Author response: Thank you for your comments. We have added the following inclusion/exclusion criteria:

Inclusion criteria:

- Adults provided with hearing aids for the primary complain of hearing loss
- Any healthcare practitioners, for example audiologists, supporting adults provided with hearing aids for hearing loss
- Communication partners, for example family members, supporting adults provided with hearing aids for hearing loss
- Adult audiology patient pathway including assessment and auditory rehabilitation
- Any study design. Documents such as editorials, opinion pieces, commentaries, process evaluations, qualitative research, programme manuals and systematic reviews may be included, if holding information relevant to developing a programme theory.
- Any outcomes related to the use of hearing aids by adults with hearing loss

Exclusion criteria:

- Participants < 18 years old
- Documents which do not summarise an empirical study
- Non-English papers

10. Why is that, given that HA: s have undergone a tremendous development during the last 10-15 yrs?

Author response: Thank you for this comment. The review team have discussed this and have decided to keep the searches with no date restriction. We acknowledge that HAs have undergone tremendous development during the last decade or so, however as our review is encompassing the whole adult audiology pathway, not just the technological aspects of HAs, we feel that by limiting the included documents within this time frame may mean we miss some potentially important evidence.

11. Independently?

Author response: Yes, the reviewers will conduct the screening independently. We have clarified this. Please see text below:

Selection of relevant evidence will be systematic and will follow a two-step procedure. The titles and abstracts will first be independently screened by two reviewers to assess eligibility for inclusion. Subsequently, the full texts of potentially relevant documents identified will also be independently

screened by two reviewers to assess against eligibility criteria for inclusion in the review.

12. questions also support my points above.

Author response: This is now addressed as we have added in inclusion/exclusion criteria.

13. This puzzles me a bit. I do have respect that you want to keep the review guidelines but given that you want empirical studies, to screen for study design might be important?

Author response: Realist synthesis draws on all aspects of primary study article (Jagosh, 2019) as well as background documents, grey literature, and interpretations of results by study authors. In realist synthesis evidence is not excluded unless it has no relevance to the theory area under examination. However, we have made a clear statement that documents which do not describe an empirical study will not be eligible for inclusion in the review. See the exclusion criteria listed below:

- Documents which do not summarise an empirical study

14. These have not been stated in this protocol (see comment above)

Author response: This has now been addressed as we have added inclusion/exclusion criteria.

15. Should it not be ...still indicate research on hearing aid fitting? (AR is not equal to HAfitting)

Author response: When we refer to adult aural rehabilitation, we refer to the assessment and management of hearing loss in adults. We intend this to encompass the entire audiology patient pathway, not just the hearing aid fitting appointment. We have changed the text below to refer to the whole adult audiology patient pathway.

1. Does the full-text document still indicate adult audiology patient pathway?

16. This need also to be stated in the missing eligibility criteria

Author response: This has been addressed through the addition of inclusion/exclusion criteria:

- Adults provided with hearing aids for the primary complain of hearing loss

17. Based on what you include in this section, I don't agree that this is "quality appraisal" but rather "appraisal".

Author response: Quality appraisal is a term used within realist review methodology (insert references). The RAMESES publication standards state that "Appraisal of the contribution of any section of data (within a document) should be made on two criteria: relevance and rigor." (Wong et al., 2013). We have removed the word "quality" from the subheading, but have referred to "quality appraisal" within the main body as the text as we are assessing the quality of the document with regard to relevance and rigour, which is in keeping with other published work using realist review methodology. We have added the following text to clarify:

The quality appraisal process involves assessing each document on a case-by-case basis for the concepts of relevance and rigour. The following process will be employed:

18. This is a bit vague, expand or clarify

Author response: We have added additional text to clarify:

- Rigour: whether the methods used to draw inferences are credible and trustworthy to test a particular theory in terms of sample size, data collection and analysis and inferences drawn by the authors [53]

19. It is not clear what you mean here. If you have a methodological weak study, but the paper provides “casual insights”, are the findings then reliable?

Author response: Realist review is a theory driven approach which does not adhere to the typical hierarchy of study designs (e.g. prioritising randomised controlled trials as the gold standard). Realists posit that all data sources are potentially fallible and socially constructed. By including heterogeneous study designs within the review we can develop rival theories and multiple causal pathways for different populations in different contexts. We had added a text and a key reference to highlight this approach within realist reviews:

The realist approach does not adhere to the hierarchy of study designs used in other types of reviews e.g. systematic reviews. In realist review, diverse data sources will be used to further our understanding of how and why programmes function [54].

20. I suggest remove. The term “quality” here lead one’s mind towards quality of the included studied.

Author response: We have removed the word “quality” from the subheading, but have referred to “quality appraisal” within the main body as the text as we are assessing the quality of the document with regard to relevance and rigour, which is in keeping with other published work using realist synthesis methodology. See text below:

The quality appraisal process involves assessing each document on a case-by-case basis for the concepts of relevance and rigour. The following process will be employed:

21. It’s does not clear why you do this. Probably to enhance consistency in the extraction process, or am I wrong? Please, clarify this statement.

Author response: Thank you. Yes, text added to clarify:

Any disagreements or ambiguity in relevance or rigour will be resolved by discussion with the wider review team to enhance validity and consistency.

22. Perhaps elaborate a bit about the pilot. Generally, in a literature review, this part is essential.

Author response: We have included additional text to clarify how we will pilot and modify the data extraction form. Please see additional text below:

The bespoke data extraction form will be developed, piloted with a small diverse sample of included documents (e.g. 3, representing peer-reviewed and grey literature), and used to record pertinent

characteristics of each included document. As the review progresses and the initial programme theories are iteratively refined, the data extraction form will be modified to ensure all new insights which are relevant to the programme theories are captured. As the data extraction form is modified, included documents will be revisited to ensure that all appropriate and relevant data is extracted.

23. This statement also deserves a more in-depth explanation. It makes sense why (and how) you inductively can identify codes. It's harder to grasp why (and how) the deductive approach will be applied.

Author response: A deductive approach will be applied to the data based on the initial programme theories developed in step 1 of the review. We have added the following text to clarify:

Coding of the data will be both inductive (i.e., created during analyses of the data) and deductive (i.e. created from the initial programme theories, identifying data that either confirms or refutes the theory under examination); both approaches will be used to identify contexts, mechanisms and outcomes, and the associations between them.

24. No 5. What exactly do you mean here? It's does not clear how you will conduct the testing. And what does "additional documents" really mean?

Author response: We have added the following additional text to clarify:

5. As the synthesis progresses, it may be necessary to conduct additional iterative literature searches, in order to examine or explore particular aspects of the refined programme theory. Any additional documents identified, will provide evidence to support, refute or refine the programme theory [49].

25. Where will you find these people? Have you already asked them to participate? If yes, have they agreed to participate and how will you consult them? These issues are important to address from a feasibility point of view.

Author response: We are currently working with key stakeholders on our larger body of work. These stakeholders were identified through professional contacts of the research team. We have added text below to clarify:

Stakeholders, identified through professional contacts of the research team, including healthcare professionals, people with hearing loss and those who they communicate regularly with will be consulted and will assist in the refinement of the final programme theory through workshop discussions [56].

Reviewer 3 comments

1. My only remaining question relates to the planned schedule of this process. There are a number of iterations described in the protocol which apparently render accurate time estimates difficult. Nevertheless, it would be interesting for the readers to know approximately which timeline and overall effort would be required. When can results of the study be expected?

Author response: As realist reviews are an iterative process, we cannot state which stage of the review will happen as typically you move back and forth through each stage of the review. However, we have added in text to clarify when the review started and the anticipated end date. Please see text

below:

The review protocol is registered with PROSPERO (282049). The review started in September 2021 and the anticipated end-date is no later than end-May 2023.

VERSION 2 – REVIEW

REVIEWER	Granberg, Sarah Univ Orebro, School of health sciences
REVIEW RETURNED	18-May-2022
GENERAL COMMENTS	Thank your for responding so well to reviewers comments. This is a very nice protocol and I have no further comments that needs to be addressed. Looking forward to reading about the study once it is completed and published.